

# Warming drove the Expansion of Marine Anoxia in the Equatorial Atlantic during the Cenomanian Leading up to Oceanic Anoxic Event 2

Mohd Al Farid Abraham[1,2], Bernhard David A. Naafs[1], Vittoria Lauretano[1], Fotis Sgouridis[3], Richard D. Pancost[1]

[1]Organic Geochemistry Unit, School of Chemistry and School of Earth Sciences, University of Bristol, BS8 1TS, United Kingdom
[2]Geology Department, Faculty of Science and Natural Resources, Universiti Malaysia Sabah, Jalan UMS, 88400, Kota Kinabalu, Sabah, Malaysia
[3]School of Geographical Sciences, University of Bristol, BS8 1SS, United Kingdom

Correspondence to: al.farid@ums.edu.my

**Abstract.** Oceanic Anoxic Event (OAE) 2 (~93.5 millions of years ago) is characterized by widespread marine anoxia and
elevated burial rates of organic matter. However, the factors that led to this widespread marine deoxygenation and the possible link with climatic change remain debated. Here, we report long-term biomarker records of water column anoxia, water column and photic zone euxinia (PZE), and sea surface temperature (SST) from Demerara Rise in the equatorial Atlantic that span 3.8 million years of the late Cenomanian to Turonian, including OAE 2. We find that total organic carbon (TOC) contents are high but variable (0.41-17 wt. %) across the Cenomanian and increase with time. This long-term TOC increase coincides with a
TEX$_{86}$-derived SST increase from ~ 35 to 40 °C as well as the episodic occurrence of 28,30-dinorhopane (DNH) and lycopane, indicating warming and expansion of the oxygen minimum zone (OMZ) predating OAE 2. Water column euxinia persisted through much of the late Cenomanian, as indicated by the presence of C$_{35}$ hopanoid thiophene, but only reached the photic zone during OAE 2, as indicated by the presence of isorenieratane. Using these biomarker records, we suggest that water column anoxia and euxinia in the equatorial Atlantic preceded OAE 2 and this deoxygenation was driven by global warming.

## 1 Introduction

Ocean Anoxic Event (OAE) 2, which occurred at the Cenomanian-Turonian Boundary (93.5 Ma), is the last major Cretaceous anoxic event (Jenkyns, 2010) and lasted around 430 to 700 thousand years (Voigt et al., 2008; Eldrett et al., 2015; Meyers et al., 2012). It is characterized by a global decline in ocean oxygenation and widespread burial of black shales rich in organic matter (OM) (Jenkyns, 2010; Schlanger and Jenkyns, 1976). Additional evidence for the enhanced burial of $^{13}$C-depleted OM
comes from the globally recorded positive stable carbon isotope (δ$^{13}$C) excursion across OAE 2 (Erbacher et al., 2005; Sinninghe Damsté et al., 2010; Takashima et al., 2010; Schlanger et al., 1987; Jarvis et al., 2006, 2011). Notably, carbon burial



rates and the magnitude of the positive carbon isotope excursion (CIE) vary among regions, with southern North Atlantic sites, for example, characterized by particularly high organic matter contents, with total organic carbon (TOC) contents of 50 % (Monteiro et al., 2012 and references therein) and CIEs up to 6 ‰ (Erbacher et al., 2005; Arthur et al., 1988).

For more than 40 years (Schlanger and Jenkyns, 1976), the causal mechanisms for OAE 2 have remained contested, but the leading hypothesis is that a large input of volcanically sourced carbon dioxide into the atmosphere (Barclay et al., 2010), associated with the emplacement of the Caribbean Large Igneous Province (CLIP; Snow et al., 2005) and the High Arctic Large Igneous Province (HALIP; Schröder-Adams et al., 2019), increased global temperatures. Subsequent feedback mechanisms, such as an increase in continental weathering (Pogge Von Strandmann et al., 2013), led to an enhanced ocean
nutrient budget that fuelled high productivity regimes that were further supported by ocean upwelling (Lüning et al., 2004). These drove widespread marine deoxygenation and led to higher organic carbon burial rates across the world (Jenkyns, 2010; Monteiro et al., 2012). Potentially, as much as 50 % of the ocean volume was deoxygenated during OAE 2 based on model experiments (Monteiro et al., 2012), although other approaches yield lower estimates (Clarkson et al., 2018). Regardless, there is strong evidence for widespread marine deoxygenation, which impacted key-biogeochemical cycles (Naafs et al., 2019).

However, these mechanisms are dependant to various degrees on pre-conditioning and the background state of the mid-Cretaceous Ocean and climate. A compilation of sea surface temperature (SST) across the Cretaceous shows that the Cenomanian was characterized by the highest values of the Cretaceous with tropical sites reaching temperatures over 35 °C (O'Brien et al., 2017), but the detailed evolution of SSTs remains poorly constrained.  Changes in organic burial rates in the proto–North Atlantic Ocean, both during OAE 2 and preceding it, could have been caused by these high temperatures;
alternatively, they could highlight the role of marine gateways in controlling the incursion of oxic or anoxic water masses that induced widespread marine anoxia (Laugié et al., 2021). Scaife et al. (2017) suggested that the mid-Cenomanian Event (MCE; 96.49 Ma; Batenburg et al., 2016) was a prelude to the onset of the OAE 2, characterized by mercury evidence for subaerial LIP emplacement and a positive CIE of ~1 ‰ (Jarvis et al., 2006; Joo and Sageman, 2014; Joo et al., 2020).

Here, we explore the detailed Cenomanian evolution of marine anoxia and its link with SSTs at Ocean Drilling
Programme (ODP) Leg 207 Demerara Rise in the equatorial North Atlantic Ocean. ODP Leg 207 comprises five sites (Site 1257 to 1261) that recovered sediments ranging from Albian to Pleistocene age (Erbacher et al., 2004). Notably, the occurrence of marine anoxia and photic zone euxinia in this basin has been previously reported from proximal Site 1260 using the biomarker lycopane and trace metals that increase in abundance before and during OAE 2 (van Bentum et al., 2009). However, that study only reported the latest part of the Cenomanian prior to OAE 2.

The sediment from the more distal and deeper Site 1258 could provide an extended Cenomanian succession and a long-term paleoenvironmental record of the late Cenomanian. We determined the occurrence of water-column anoxia using the biomarker 28,30-dinorhopane (Moldowan et al., 1984). Water column anoxia was also reconstructed using lycopane as a proxy for the oxygen minimum zone (OMZ; Sinninghe Damsté et al., 2003; Adam et al., 2006), complementing the published record from Site 1260 (van Bentum et al., 2009). Additionally, we reconstruct water column euxinia (sulfidic condition) based
on the occurrence of $C_{35}$ hopanoid thiophenes (Valisolalao et al., 1984; Sinninghe Damsté et al., 1995). The expansion of





euxinic conditions into the photic zone was reconstructed by the abundance of the biomarker isorenieratane (Sinninghe Damsté et al., 2001), extending the record from Site 1260 (van Bentum et al., 2009). In parallel, we reconstructed SST at Site 1258 based on the membrane lipids (isoGDGTs) of Thaumarchaeota – TEX$_{86}$ (TetraEther indeX of 86 carbons; Schouten et al., 2002; Kim et al., 2010) – expanding on the previously published low-resolution data from Site 1258 (Forster et al., 2007). Ultimately, we link this high-resolution record of water column anoxia and euxinia with the evolution of climate (e.g., SST) during the Cenomanian and test the hypothesis that warming drove ocean deoxygenation during the 3.7 million years preceding OAE 2.

## 2 Site Location

Ocean Drilling Programme (ODP) Leg 207, Site 1258 of Demerara Rise is a deep-water site at 3192.2 meters below sea level on the continental shelf north of Suriname in the equatorial Atlantic. During the Cenomanian this site was located at a latitude of ~5 °N (Figure 3.1). This study investigated 123 sediments from Site 1258 (hole B) that was cored to 460.9 meters below sea floor with sediment recovery of 76.3 % and spanning the Cenomanian to Turonian. The lithostratigraphic units for the Cenomanian to Turonian were defined as Unit IV- black finely laminated calcareous claystone with relatively high organic matter contents. It is preceded by Albian phosphoritic calcareous claystone (Unit V) and followed by younger Units (III to I) that span the Campanian to Miocene and mainly comprise calcareous and siliceous microfossils and clay (Erbacher et al., 2004).

Total organic carbon content at Site 1258 increases from the Albian to Cenomanian-Turonian Boundary (CTB) with a maximum of ~28 wt. % during OAE 2; much lower TOC contents are found following the CTB. Carbonate content ranges from 30 % to 80 %, with lowest values (~5 %) occurring in OAE 2 sediments. The OAE 2 itself is identified at Site 1258 based on a positive excursion in $\delta^{13}C_{org}$ values by ~6 ‰, consistent with previous studies and global change in the C-cycle (Sageman et al., 2006; Li et al., 2017). Limited carbonate preservation has hindered the effort to constrain $\delta^{13}C_{carb}$ across OAE 2.

Due to extensive prior sampling, relatively few sediments remain from the OAE 2 interval, and here we predominantly focus on the long-term trends during the Cenomanian leading up to this event. The age-depth model for the Cenomanian are based on published data (Friedrich et al., 2008) and three tie points: a) the Middle Cenomanian event (95.7 Ma); b) the last occurrence of the nannofossil marker *Corollithion kennedyi* (94.1 Ma); and c) the onset of the OAE 2 positive carbon excursion (~300 kyr prior to CTB; 93.8 Ma; Erbacher et al., 2005). The interval of OAE 2 at Site 1258 (422 to 426 m composite depth) is estimated to have lasted for 550 kyr (Meyers et al., 2012).



## 3 Materials and Methods

The stable carbon isotopic composition of bulk organic matter ($\delta^{13}C_{org}$; expressed relative to Vienna PeeDee Belemnite) and total organic carbon (TOC; wt. %) contents at Site 1258 were analysed on aliquots (5-10 mg) of homogenised black shale samples using an Elemental Analyser (EA) coupled with an Elementar Isoprime Precision (IRMS), following carbonate removal from the samples via acidification as described by Hedges and Stern (1984). Analyses were carried out in duplicates with the average reported here (standard deviation < 0.3). The instrument was normalized using organic reference materials of

USGS61 (–35.05 ± 0.04 ‰), USGS62 (–14.79 ± 0.04 ‰), and USGS63 (–1.17 ± 0.04 ‰), as reported by Schimmelmann et al. (2016). These new $\delta^{13}C$ and TOC data were combined with published data from Site 1258 (Erbacher et al., 2005; Friedrich et al., 2008).

For biomarker characterization, we extracted 5 g each of 123 ground samples with 15 ml of a dichloromethane (DCM):methanol (MeOH; 9:1, v/v) azeotrope using an ETHOX EX microwave extraction system. 2500 ng of 5α-Androstane

were added as an internal standard prior to extraction. The total lipid extract (TLE) was separated via column chromatography into apolar and polar fractions with 4 ml of hexane: DCM (9:1, v/v) and 3 ml of DCM: MeOH (1:2, v/v), respectively. The apolar fraction, containing the anoxia and euxinia biomarkers, was analysed using a Thermo Scientific™ ISQ Series Single Quadruple gas chromatography-mass spectrometer (GC-MS). The separation of compounds was performed on a Zebron non-polar column (50 m x 0.32 mm, 0.10 μm film thickness). The injection volume was 1 μl, and the GC was programmed for

injection at 70 °C (1 min hold), heating to 130 °C at a rate of 20 °C/min, then to 300 °C at 4 °C/min, followed by a 24 min hold. The carrier gas was helium, with a flow rate of 3 ml/min. The GC-MS continually scanned between $m/z$ 50 to 650. It is operated in EI-mode at 70 eV at ion source temperature of 200°C, and the interface temperature between GC and MS was maintained at 300 °C. To monitor instrument stability, a fatty acid methyl ester standard mixture was injected daily.

The concentration of biomarkers was determined by integrating the peak on a partial mass chromatogram ($m/z$) of

known fragments ion of biomarkers relative to the peak area of the standard on similar $m/z$ trace. Due to the variety of response factors, we do not convert these to true concentrations. The biomarkers were identified based on published spectra, characteristic mass fragments and retention times. Briefly, the $C_{28}$ 28,30 dinorhopane (DNH) that serves as proxy for water column anoxia was identified based on $m/z$ 191, 163 and 384 fragments (Moldowan et al., 1984). Lycopane, which indicates the presence of an oxygen minimum zone, was identified based on $m/z$ 71, 113, 183, 253, 309, 337, 407, 477; M+ = 563), but

it co-elutes with the $C_{35}$ $n$-alkane (Sinninghe Damsté et al., 2003). The incorporation of sulfur into biomarkers indicates water column euxinia and is traced using the $C_{35}$ hopanoid thiophene, identified from its $m/z$ 191, 369 and 97 (Valisolalao et al., 1984). Photic zone euxinia (PZE) was reconstructed based on the biomarker isorenieratane, $C_{40}$ compounds with characteristic fragments of $m/z$ 133, 134 and M+ 546 (Koopmans et al., 1996).

The polar fractions containing the glycerol dialkyl glycerol tetraethers (GDGTs) were dissolved in

hexane/isopropanol (99:1, v/v) and passed through 0.45 μm polytetrafluoroethylene filters prior to Single Ion Monitoring (SIM) analysis on a ThermoFisher Scientific Accela Quantum Access triple quadrupole mass spectrometer coupled to a high-





performance liquid chromatography-mass spectrometry (HPLC-MS) system. The LC instrument methods followed Hopmans et al. (2016). To reconstruct $TEX_{86}$-based SST (Schouten et al., 2002), we evaluated secondary influences on $TEX_{86}$ using established GDGT indices such as the Branched Isoprenoidal Tetraether Index (BIT Index) to preclude excessive soil input

(Hopmans et al., 2004); percentage of GDGT-0 (Sinninghe Damsté et al., 2012) to evaluate potential contributions from methanogenic archaea; the Methane Index (Zhang et al., 2011) to preclude contributions from methanotrophic Euryarchaeota; and the GDGT-2/GDGT-3 ratio that distinguishes the contribution of deep-marine (high ratio) versus shallow subsurface (low ratio) ammonia-oxidising Thaumarchaeota (Taylor et al., 2013). Then, $TEX_{86}$-based SSTs  were determined using Bayesian Spatially varying Regression (BAYSPAR) with a prior of $30 \pm 20$ °C and search tolerance of 3 standard deviations, using

MATLAB (Tierney and Tingley, 2014). We combined our higher-resolution data with previously published $TEX_{86}$ records from Site 1258 (Forster et al., 2007), converting those to SST using the same BAYSPAR methodology.

## 4 Results

The long-term $\delta^{13}C_{org}$ record, based on a combination of data from this study and published data (Erbacher et al., 2005; Friedrich et al., 2008), is relatively stable throughout most of the Cenomanian (97 to 93.8 Ma), ranging from -30 to -27 ‰ and

increasing slightly through the Cenomanian (Figure 3.3A). A major positive excursion up to maximum values of ~ -21 ‰ marks the OAE 2 interval between 93.8 to 93.5 Ma (422 to 426 m). TOC contents vary dramatically but gradually increase from 1 to 17 wt. % in pre-OAE Cenomanian sediments and reach their highest values of 28 wt. % during OAE 2 (Figure 3.3B).

$TEX_{86}$-based SSTs, based on the BAYSPAR calibration of Tierney and Tingley (2014), decrease slightly during the early Cenomanian from an average of ~34 °C to a minimum of ~32 °C (95.73 Ma) in the mid-Cenomanian, coinciding with

the Mid-Cenomanian positive carbon isotope Excursion (MCE; Figure 3.3C). The SST then exhibits a significant long-term – but episodic – increase, reaching a maximum of ~43 °C at around 93 Ma. Our reported SSTs are lower than those of Forster et al. (2007), likely due to interlaboratory variations in LC-MS conditions and modified LC-MS analytical protocol (Schouten et al., 2013).  There is no evidence for secondary influences on isoprenoidal GDGT distributions that would preclude their use in SST estimation. The Cenomanian average for  the BIT Index is 0.1 (Hopmans et al., 2004; Weijers et al., 2006) and the

Methane Index is 0.2 (Zhang et al., 2011), both of which are low (Supplementary Material Table 2). GDGT-2/GDGT-3 ratios have been used to explore the balance of shallow vs deep-dwelling Thaumarchaeota inputs (Taylor et al., 2013). Values here are low (average 2.2), suggesting that the isoprenoidal GDGTs are predominantly derived from the shallow water ammonia-oxidising Thaumarchaeota community, and they are consistent with GDGT-2/GDGT-3 values throughout the Mesozoic (average of 2.6) (Rattanasriampaipong et al., 2022).  These values are lower than those in modern oceans and it remains unclear

if this affects reconstructed SSTs (Rattanasriampaipong et al., 2022), but the lack of any long-term change in GDGT-2/GDGT-3 ratios in Cenomanian Demarara Rise sediments indicates that secular trends are robust.

The relative abundance of dinorhopane (DNH; abundance relative to total hopanes; Figure 3.3D), a biomarker indicative of water column anoxia (Peters et al., 2004), is low in the early Cenomanian, exhibits multiple maxima in the middle



and late Cenomanian sediments, but is again low during OAE 2. The lycopane index (Figure 3.3E), also indicative of water column anoxia and/or an expanded oxygen minimum zone (Sinninghe Damsté et al., 2003), closely tracks the DNH relative abundance ($r^2 = 0.67$, Figure 4). The lycopane index is low in the lowermost part of the section, but from the mid-Cenomanian until OAE 2 it is highly variable with at least eight maxima and values up to 35 (95.17 Ma). Intriguingly, lycopane indices are relatively low during OAE 2, and this is in agreement with the previously published data from proximal Site 1260 of Demerara Rise (van Bentum et al., 2009). Low lycopane and DNH indices from OAE 2 could partially reflect their reaction with hydrogen sulfide and incorporation into a S-bound pool of OM (Sinninghe Damsté et al., 2014), and this is discussed below.

The $C_{35}$ hopanoid thiophene concentrations (Sinninghe Damsté et al., 1990) are low or below detection in early Cenomanian sediments, suggesting minimal water column euxinia. However, concentrations increase from the mid-Cenomanian towards OAE 2 (Figure 3.3F). Isorenieratane, derived from the green sulfur bacteria carotenoid isorenieratene (French et al., 2015 and references therein) and therefore a biomarker for PZE (Sinninghe Damsté et al., 2001), occurs in only two samples, both from the OAE 2 interval, although the sampling resolution for OAE 2 was limited. Crucially, isorenieratane could be partially sequestered in the S-bound fraction of organic matter (Sinninghe Damsté and Köster, 1998; Ma et al., 2021). van Bentum et al. (2009) investigated the sulfur-bound biomarkers and reported the occurrence of isorenieratane only during the OAE 2 onset at Demerara Rise (Site 1260) with no signal prior to the event.

## 5 Discussions

### 5.1 Marine anoxia expansion during Cenomanian

The relatively high TOC contents during the Cenomanian suggest that these black shales at Site 1258 were deposited under the influence of bottom-water oxygen limitation (Burdige, 2007). Following OAE 2, TOC contents decrease to values < 1 wt. % (Erbacher et al., 2004), remaining low throughout the late Cretaceous and Cenozoic, including during other prolonged and transient greenhouse climates (Frieling et al., 2018). This suggests that these anoxic conditions, driven by high organic matter burial rates at Demerara Rise during the mid-Cretaceous, were facilitated by basin geometry during the early opening phases of the South Atlantic (Friedrich and Erbacher, 2006; Donnadieu et al., 2016). However, Cenomanian TOC contents also vary on both short and long- timescales, the latter most evident in an increase in average TOC contents from the Albian through the Cenomanian and culminating in OAE 2 (up to 28 wt. %), suggesting that basin geometry is not the only factor governing organic matter burial rates.

At the base of the studied interval, TOC contents range from 1 to 6 % and indicate that bottom water suboxic conditions could have been present even during deposition of the lowermost sections from the early Cenomanian (Arthur et al., 1987; Trabucho Alexandre et al., 2010; Berrocoso et al., 2010) and possibly the Albian. TOC contents in excess of 5 % become common in the mid-Cenomanian, alongside black shale lamination and the absence of benthic bioturbation (Erbacher et al., 2003); those features and the concomitant decrease in the abundances and diversity of foraminifera (Friedrich et al., 2008), indicate bottom water anoxia. Then, in the lead-up to OAE 2, the TOC increases up to 17 wt. %, similar to the high





TOC of ~19 wt. % that occurs just before the onset of OAE 2 at Site 367, Cape Verde Basin (Sinninghe Damsté et al., 2008), which is located at the conjugate margin to the east.

As TOC contents increase in the mid- to late-Cenomanian, so do the DNH relative abundances and lycopane index. The sediments with elevated DNH proportions are exceptional in the geological record. In our samples, DNH is sometimes

the most abundant hopane and even one of the dominant compounds in the apolar fraction; this is rare in rocks of any age (Słowakiewicz et al., 2015) and has been linked to the persistence of a strong OMZ, such as during the Monterey Event in the Miocene (Sinninghe Damsté et al., 2014). Those same DNH-rich horizons have very high lycopane indices, similar to those associated with strong OMZs in today's oceans, including the Black Sea (Sinninghe Damsté et al., 2003). The expansion of anoxia through the water column at Demerara Rise also has been invoked by the low enrichment factor (EF ~1) of manganese

during most of the Cenomanian (van Bentum et al., 2009), attributed to the dissolution of $Mn^{2+}$ and its mobilisation into an expanded OMZ (Hetzel et al., 2005). The decline in benthic foraminifera assemblages from the early to late Cenomanian provides further evidence for oxygen depletion within the water column (Friedrich et al., 2009). Together, our DNH and lycopane results build on the low-resolution lycopane record of van Bentum et al. (2009) and indicate a long-term increase in water column anoxia mediated by shorter-term variations.

Intriguingly, both the lycopane and DNH indices are low during the OAE 2 interval. This has also been reported for lycopane index at nearby Site 1260 by van Bentum et al. (2009), although their record did not extend far into the Cenomanian. Although there is great spatial variability in OAE 2 conditions (Jenkyns, 2010), the presence of isorenieratane and very high TOC contents at Sites 1258 and 1260 (van Bentum et al., 2009 and this work) indicate that the most extreme water column anoxia (and euxinia) at Demerara Rise occurred during the OAE 2 interval. If the lycopane index is driven by its selective

preservation relative to terrestrial *n*-alkanes (Sinninghe Damsté et al., 2003), then we would expect it to also be highest during OAE 2. Instead, we argue that the low values of both lycopane and DNH indices during OAE 2 are driven by a further expansion of anoxia that favoured other microorganisms at the expense of the lycopane and DNH-producers. In particular, the DNH and lycopane producers, possibly chemoautotrophs living at redox boundaries of a strong OMZ, are replaced during OAE 2 by green sulfur bacteria thriving under euxinic conditions. The co-occurrence of high concentrations of isorenieratane

and DNH is uncommon (Słowakiewicz et al., 2015), suggesting that the respective source organisms require specific and distinct oceanographic conditions. Recent studies suggest that DNH is a diagenetic product of $C_{28}$ 28,30-dinorhopene (Sinninghe Damsté et al., 2014), with both the product and precursor indicating a stratified palaeowater column. Sulfidic conditions could have contributed to the low measured abundances of lycopane and DNH during OAE 2, as their unsaturated precursors are also prone to sulfurization. However, their abundances do not decrease when water column euxinia (but not

PZE) becomes widespread (see below), and we note that Sinninghe Damsté et al. (2014) argued for rapid diagenetic conversion of $C_{28}$-dinorhopene (potential precursor) into DNH and aromatic hopanoids that are 'shielded' from reactions with sulfide.

Although variations in $C_{35}$-hopanoid thiophene concentrations do not match those of lycopane indices nor DNH abundances, they do provide evidence for a long-term increase in excess free inorganic sulfide in the water column through the Cenozoic and especially into OAE 2 (Figure 3F). In particular, Sinninghe Damsté et al. (1990) argued that abundant S-





bound OM was evidence for water column euxinia, where OM could compete favourably for reduced sulfur due to the limited availability of reactive iron (Fe). This also gives rise to the coupling of the S and OC cycles, with sulfurization facilitating OM burial (Werne et al., 2004; Raven et al., 2018) while removing S from the oceans.

Our work adds to inorganic geochemical studies that also argued for a progressive deoxygenation of the southern North Atlantic leading up to OAE 2. For example, a time lag of 75 kyr has been estimated for the dramatic drawdown of ocean

vanadium (V; a proxy for water column anoxia) during the late Cenomanian and that of molybdenum (Mo; a proxy for water column euxinia) after the onset of OAE 2 (Owens et al., 2016; Figure 3.3). Ostrander et al. (2017) indicated a shorter lag of 43 kyr between the deoxygenation of the water column and the widespread carbon burial of OAE 2 using thallium isotopes (Tl) linked to manganese oxide burial. Collectively, both studies indicate progressive deoxygenation prior to and into OAE 2. Our biomarker records, although limited for OAE 2 itself, build on these by confirming that the expansion of water column anoxia

preceded the PZE during OAE 2 and adding new evidence that the expansion of water column anoxia in the central Atlantic started as early as the MCE.

## 5.2 TEX$_{86}$ sea surface temperature estimates track marine anoxia during the Cenomanian

The prolonged deposition of organic black shales at Demerara Rise was likely facilitated by a combination of restricted palaeogeography that allowed nutrient trapping to maintain high primary productivity and enhanced preservation due to the

lack of deep-water ventilation (Trabucho Alexandre et al., 2010). The Demerara region is proximal to the nearly-closed Equatorial Atlantic Gateway (EAG) and could have acted as a 'nutrient trap' due to dynamic estuarine circulation between southwest flowing Tethyan waters and Pacific waters via the Central American Seaway (CAS; Berrocoso et al., 2010; Topper et al., 2011; Trabucho Alexandre et al., 2010). However, model simulations with a shallow-depth CAS configuration imply that marine anoxia within the Atlantic Ocean remains stable even without estuarine circulation (Laugié et al., 2021). This

indicates an additional causal mechanism of prolonged marine anoxia, which is likely to be associated with the Cenomanian climatic condition. Our biomarker data also indicate an important role for additional, potentially climatic mechanisms by showing that water column anoxia was not constant during Cenomanian times but progressively expanded upward into the water column.

Our TEX$_{86}$-derived SSTs (new data combined with the previously published data of Forster et al., 2007) show an

early Cenomanian cooling period followed by an increase of SST from the mid-Cenomanian towards OAE 2. Notably, this gradual increase in SST up to 43°C ± 3.5°C coincided with the deoxygenation of the ocean in this region, from water-column anoxia to water-column euxinia, and ultimately photic zone euxinia as indicated by the appearance of DNH, lycopane, C$_{35}$ hopanoid thiophene and isorenieratane, respectively. Therefore, this extends the occurrence of marine water column anoxia predating OAE 2 to the post-MCE late Cenomanian and directly links its expansion to SST, at least for this site (Figure 3.3).

The Demerara region was likely bathed by warm saline intermediate water as a result of warm surface water at mid- to high-latitudes that propagate via deep-water circulation (Friedrich et al., 2008). Therefore, we argue that the expansion of water anoxia and the oxygen minimum zone (based on our DNH and lycopane indices) is linked to the displacement of warm





saline Demerara Bottom Water (DBW) which is overridden by southwest-flowing Tethyan waters (Berrocoso et al., 2010). This mass water displacement is evidenced by sharp transitions in neodymium isotopes, with Tethyan Waters having a heavier

value that is only recorded in shallow water (Site 1260), in contrast with the lighter values at Site 1258 that characterise DBW (Berrocoso et al., 2010). Crucially, the upper boundary of the warm saline DBW (Friedrich et al., 2008) likely fluctuated due to the high eustatic sea level associated with thermal expansion (Haq, 2014). Hence, it is probable that temperature-controlled ocean circulation and sea level sustained and controlled the Cenomanian black shale deposition through a combination of nutrient-rich and oxygen-poor deep-convection, recycling of benthic phosphorus (Van Cappellen and Ingall, 1994; Mort et al.,

2007), elevated nutrient inputs caused by warming-induced continental weathering (Monteiro et al., 2012; Nana Yobo et al., 2022) and high surface productivity. Collectively, these mechanisms suggest that water column anoxia at the southern margin of the North Atlantic during OAE 2 but also during the Cenomanian was governed by paleogeographic configuration but modulated by long-term climate change such as temperature (SST). Most likely, this Cenomanian warming was global as it is also seen in the global compilation (O'Brien et al., 2017) and driven by volcanism-induced increases in $CO_2$ (Barclay et al.,

270 2010).

To explore the partial pressure of atmospheric carbon dioxide ($pCO_2$) during Cenomanian, we also determined the $\delta^{13}C$ values of the marine photoautotroph biomarker phytane (see Supplementary Information). The $\delta^{13}C$ values of phytane are low (among the lowest of the Phanerozoic), confirming high $pCO_2$ during Cenomanian (Supplementary Information; Table 3). Phytane $\delta^{13}C$ values are also rather stable, but that is likely due to high $pCO_2$ where carbon isotope fractionation is saturated

(e.g., Pancost et al., 2013) rather than a lack of $pCO_2$ change. Due to the lack of carbonate for most of our samples, we cannot rigorously determine carbon isotope fractionation and therefore quantify $pCO_2$. Given the SST change, it is likely that $pCO_2$ increased, but alternatively warming could have been locally amplified by an equatorward shrinkage of the Hadley circulation, causing atmospheric heat to be preserved within the equatorial region and promoting tropical warmth (Hasegawa et al., 2012). During OAE 2, phytane $\delta^{13}C$ increased dramatically, very likely indicating a $pCO_2$ decrease and a negative feedback on global

warming via widespread organic carbon burial as extensively discussed elsewhere (e.g., Sinninghe Damsté et al., 2008).

Although inferred $pCO_2$ rise and SST warming appear closely linked to the expansion of anoxia during the Cenomanian, it was likely not the primary driver for long-term anoxia in the basin. The abrupt termination of OAE 2 and the associated decline in TOC contents, lycopane and DNH indices and isorenieratane abundances occurred despite elevated SSTs that persist into the Turonian. Similarly, this persistent warming is also recorded at other sites (Robinson et al., 2019). Such

continuously high SSTs appear to be linked to elevated atmospheric $CO_2$ driven by continuous volcanic outgassing (Robinson et al., 2019) that outlasted the carbon drawdown caused by widespread organic carbon burial during OAE 2.

Regardless of the mechanism, decoupling of our SST record from redox indicators confirms that temperature is not the only driver of water-column anoxia, at least at Demerara Rise after OAE 2. We suggest that the termination of anoxic conditions at Demerara Rise is related to the exhaustion of nutrients and the collapse of elevated primary productivity (Owens

et al., 2016) or due to the tectonic opening of the EAG that reconfigured North Atlantic ocean circulation such that it no longer acted as a nutrient trap (Berrocoso et al., 2010). As such, our collective Cenomanian records document a long-term increase

in SST that caused Demerara Rise to cross several thresholds with respect to water column structure, productivity, and redox conditions. These proxies do vary throughout the Cenomanian, and future work should develop higher resolution records that could explore whether they were modulated by short-term astronomical forcing (Nederbragt et al., 2005). We suggest that

these could have been related to the nutrient status of the site, with temperature and hydrology-induced weathering delivering increased nutrients across multiple timescales.

## 6 Conclusions

We show that Demerara Rise experienced water column anoxia during the late Cenomanian leading up to the OAE 2 and that its expansion was driven by warming. Water column anoxia is evidenced by the high abundances of 28,30 dinorhopane and

lycopane, which indicate the expansion of water column anoxia and the oxygen minimum zone at Demerara Rise. The deoxygenated water column evolved into more extreme sulfidic condition during the latter part of the Cenomanian, and euxinic conditions reached the photic zone during OAE 2, as indicated by the presence of $C_{35}$ hopanoid thiophene and isorenieratane, respectively. This equatorial Atlantic evolution of marine anoxia appears to be closely linked to temperature rise, only becoming decoupled after OAE 2 and the tectonic opening of the North Atlantic, suggesting that geography was a crucial pre-

condition for the development of anoxia, but it was modulated by climatic factors.

**Author contributions.** MAFA, BDAN and RDP designed this study. MAFA performed the organic geochemical analyses. MAFA and VL generated the SST reconstructions using MATLAB. FS generated the bulk stable carbon isotopes. MAFA, BDAN, VL and RDP discussed and interpreted the data. MAFA wrote the paper, with input from all authors.

**Competing interests.** The authors declare that they have no conflict of interest.

**Acknowledgements.** We thank ERC and NEIF (www.isotopesuk.org) for funding and maintenance of the GC-MS, HPLC-MS and GC-C-IRMS instruments. The International Ocean Discovery Programme- Bremen Core Repository (IODP-BCR) MARUM supported this work through sampling assistance. MAFA is funded by the Ministry of Higher Education Malaysia and Universiti Malaysia Sabah. BDAN was funded through a Royal Society Tata University Research Fellowship.

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





550

**Figure 1: Paleogeographic location of ODP Site 1258, Demerara Rise (modern latitude: 09°27.23'N; 54°20.52'W, yellow circle), during the Cenomanian (~ 95 Ma). The map is from Scotese (2021) and shows land (green), continental shelf (light blue), deep water (dark blue) and modern territorial boundaries (solid black line). Also shown are five potentially important marine gateways; the**
555 **Equatorial Atlantic Gateway (EAG), Central America Seaway (CAS), Western Interior Seaway (WIS), East Greenland Gateway (EGS), and Tethys Seaway.**



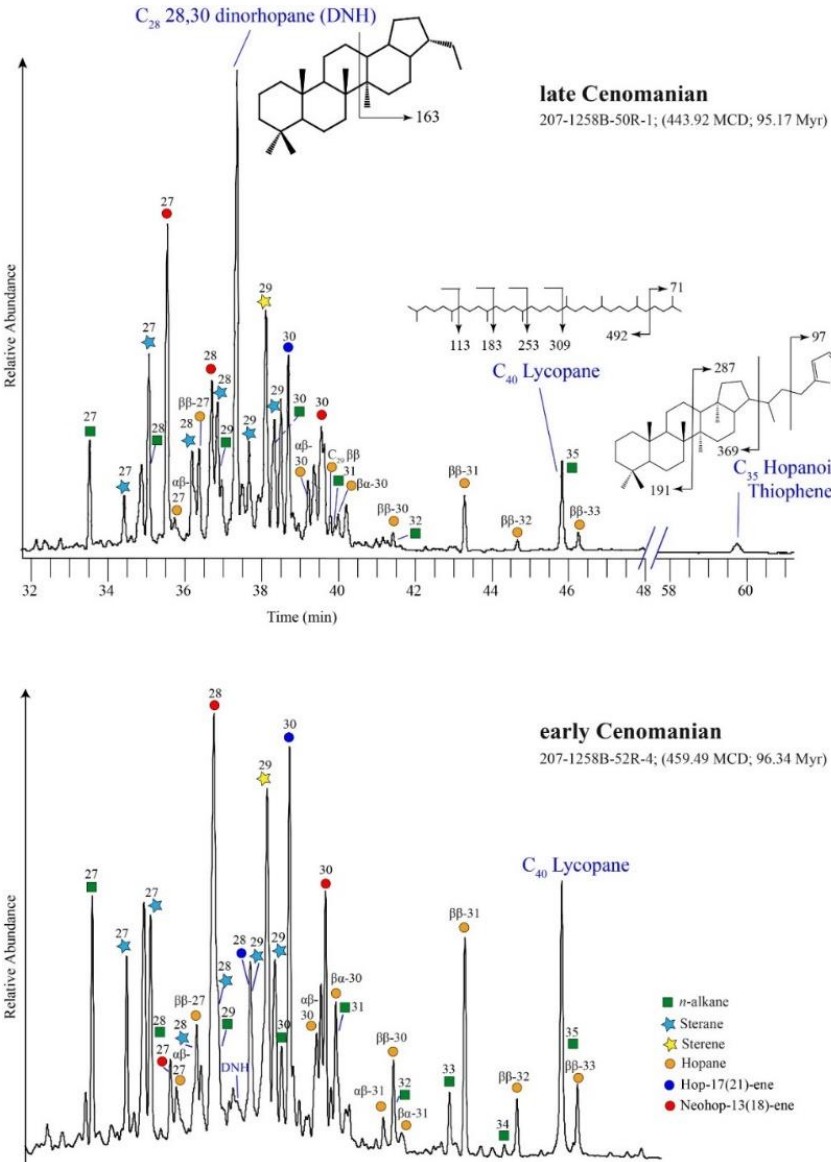

**Figure 2: Total Ions Chromatogram (TIC) of the apolar fraction of two typical Cenomanian samples at 96.34 Myr and 95.17 Myr. The partial mass structures indicate the targeted biomarkers use to determine the water column anoxia using C28 28,30-dinorhopane (m/z = 191, 163; M+ = 384); oxygen minimum zone based on lycopane (m/z 71, 113, 183, 253, 309; M+ = 492). The transition from anoxic to euxinic water column is indicated by the presence of C35 hopanoid thiophene that is absent during early the Cenomanian. The carbon number indicates from the number above the key symbols that represent suites of n-alkanes, steroids, and hopanoids.**



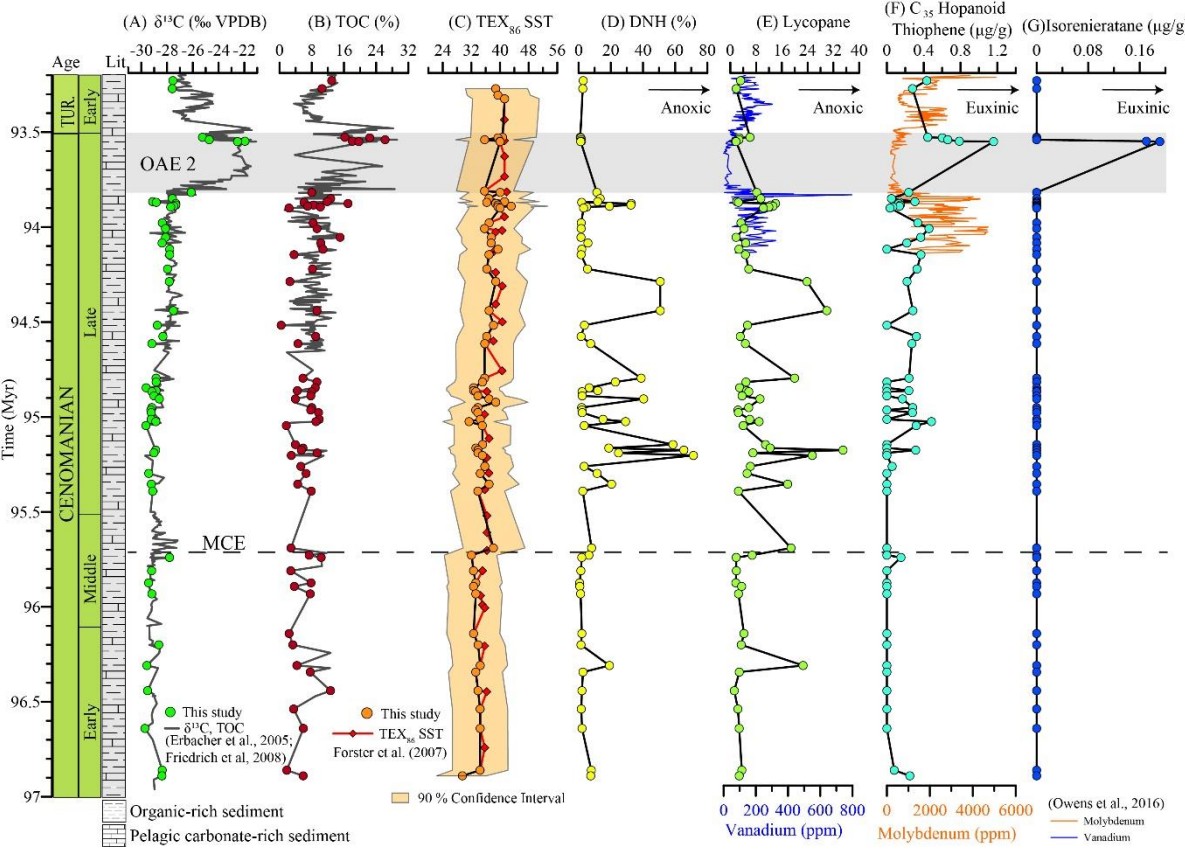

**Figure 3: Stable isotopic composition, TOC, and biomarker records for Site 1258 at Demerara Rise. From left to right, (A) Stable carbon isotopic composition of bulk organic matter, combining data from this study and published data (Erbacher et al., 2005; Friedrich et al., 2008), (B) Total Organic Carbon (TOC) content (new data and published data (Erbacher et al., 2005; Friedrich et al., 2008)), (C) BAYSPAR-calibrated TEX$_{86}$-based SST based on data of this study and Forster et al. (2007), (D) 28,30-dinorhopane/Total C$_{27}$-C$_{35}$ hopane ratio, (E) (lycopane + C$_{35}$ *n*-alkanes) /C$_{31}$ *n*-alkanes (lycopane) index, (F) concentration of C$_{35}$-hopanoid thiophenes, and (G) concentration of isorenieratane. The calibration uncertainty for BAYSPAR-calculated TEX$_{86}$-based SST is ± 3.5 °C, that is approximate to the mean (n = 123) width of 90 % confidence interval. Also shown in panels E and F (lower axes) are vanadium and molybdenum concentrations, showing their dramatic drawdown during OAE 2 (Owens et al., 2016).**



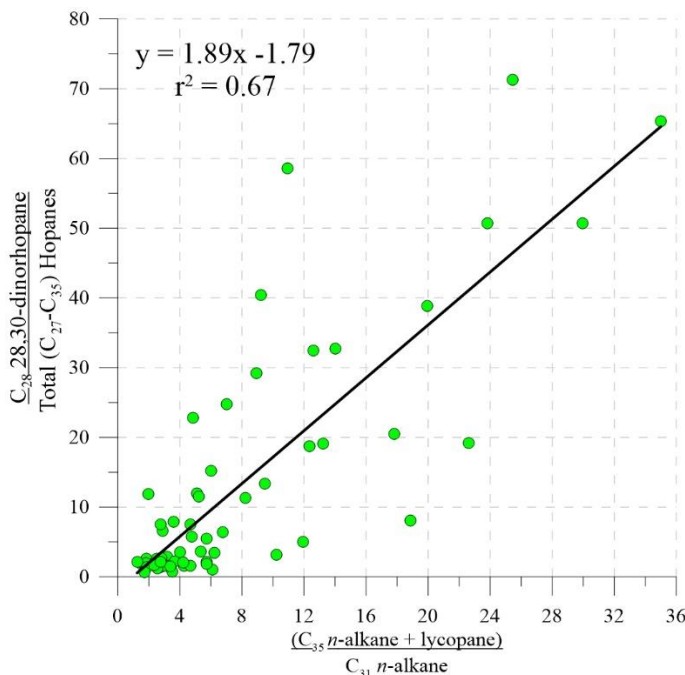

**Figure 4: Cross plot of DNH relative abundances and lycopane indices, showing the similar behaviour of these biomarkers' indicative of water column anoxia throughout the Cenomanian leading up to OAE 2.**