# Peer review of "Warming drove the Expansion of Marine Anoxia in the Equatorial Atlantic during the Cenomanian Leading up to Oceanic Anoxic Event 2"

_EGUsphere, 2023_

## Referee Comment (RC1)

**Reviews of:**

**"Warming drove the Expansion of Marine Anoxia in the Equatorial Atlantic during the Cenomanian Leading up to Oceanic Anoxic Event 2"**
egusphere-2023-260

**Vienna, April 27, 2023**

**Alexandre Pohl** (CNRS Researcher @ Biogéosciences, UMR 6282 CNRS, Université de Bourgogne, 6 Boulevard Gabriel, 21000 Dijon, France)

**Paper summary**:

Al Farid Abraham present new sea-surface temperature (SST) and organic redox proxy data for water column anoxia and water column and photic zone euxinia for the Cretaceous. Data come from Demerara Rise, a site that was situated in the central Atlantic, and more specifically from Ocean Drilling Programme Leg 207, site 1258. The data cover 3.8 Myrs, mostly preceding but also covering and slight exceeding OAE 2. The authors show that SSTs and TOC increase before OAE 2 and that water-column anoxia and euxinia spread at the studied location before OAE 2. Photic-zone euxinia only occurred during OAE 2. SST kept increasing in the aftermath of OAE 2. The authors conclude that warming played a critical role in the spread of anoxia in the central Atlantic before, and during OAE 2.

**General comment**:

The manuscript is concise and clear and overall very well constructed and pleasant to read. New data are not really unexpected nor ground-breaking (since they overlap in part with van Bentum et al., 2009 and O'Brien et al., 2017), but they are interesting and new. Above all, I really appreciated the way the authors used previous work: the reader easily follows what data were generated in this study, what data come from previous work, and the authors compare their new data with previous work in a clever and accurate way. I write this review more rapidly than usual because I am asked to send it by May 1st, and I am currently at EGU, but that's fine with me since I have only minor comments and suggest prompt publication after minor revisions anyway (Please note that I'm no geochemist and am not able to evaluate the robustness of the geochemical analyses).

**Minor comments**:

**1**. Impact of the orbital configuration: The authors very rapidly approach the question of astronomical forcing (line 294). I think they could tell a bit more about that, notably in the light of key modeling (Sarr et al., 2019) and data (Laurin et al., 2016; Li et al., 2017) studies. I think that invoking the orbit:

- is important in the introduction;
- might help explaining the onset and/or termination of OAE 2 against a background of increasing SSTs.

**2**. Please check the figures:

- I think Fig. 1 is not called in the main text, but should be kept in the MS;
- I think Fig. 2 is not called and am not sure it's really useful;
- Fig. 3 is very nice but laterally very compressed, so that all temporal trends are difficult to read (e.g., the large SST increase, which looks like a flat line; or similarly:

"TOC contents in excess of 5%" on 187: this is typically very difficult to see on the figure). Would that be possible to solve that problem? It would at least be helpful to draw vertical lines for key values (i.e., a figure 'grid'). Also, on line 141, the authors refer to some depth interval, but depth is not shown in Fig. 3. Please add it or convert depth to age in the text.

**3**. Lines 146–147: "SSTs are lower than those of Foster et al (2007)". By how much? Again, this is difficult to estimate based on Fig. 3.

**4**. Lines 182–183: "from the Albian to the Cenomanian". Albian not shown?

**5**. Lines 226–227: Regarding sulfurization of OM during OAE 2, please consider the very nice quantification by (Hülse et al., 2019).

**Technical comments**:

- Line 44: 'key biogeochemical cycles'
- Line 76 and throughout (e.g., lines 140, 142, 157): Fig. 3.1 should be Fig. 3 I think.
- Line 226: I guess "OC" stands for organic carbon, please define upon first use or remove the acronym (which I think is used here only).

**References cited**:

Hülse, D., Arndt, S., Ridgwell, A., 2019. Mitigation of Extreme Ocean Anoxic Event Conditions by Organic Matter Sulfurization. Paleoceanography and Paleoclimatology 34, 476–489. https://doi.org/10.1029/2018PA003470

Laurin, J., Meyers, S.R., Galeotti, S., Lanci, L., 2016. Frequency modulation reveals the phasing of orbital eccentricity during Cretaceous Oceanic Anoxic Event II and the Eocene hyperthermals. Earth and Planetary Science Letters 442, 143–156. https://doi.org/10.1016/j.epsl.2016.02.047

Li, Y.-X., Montañez, I.P., Liu, Z., Ma, L., 2017. Astronomical constraints on global carbon-cycle perturbation during Oceanic Anoxic Event 2 (OAE2). Earth and Planetary Science Letters 462, 35–46. https://doi.org/10.1016/j.epsl.2017.01.007

Sarr, A.C., Sepulchre, P., Husson, L., 2019. Impact of the Sunda Shelf on the Climate of the Maritime Continent. Journal of Geophysical Research: Atmospheres 124, 2574–2588. https://doi.org/10.1029/2018JD029971

---

## Author Comment (AC1)

**Responses to Reviewer #1's comments**

**Paper summary**:
Al Farid Abraham present new sea-surface temperature (SST) and organic redox proxy data for water column anoxia and water column and photic zone euxinia for the Cretaceous. Data come from Demerara Rise, a site that was situated in the central Atlantic, and more specifically from Ocean Drilling Programme Leg 207, site 1258. The data cover 3.8 Myrs, mostly preceding but also covering and slight exceeding OAE 2. The authors show that SSTs and TOC increase before OAE 2 and that water-column anoxia and euxinia spread at the studied location before OAE 2. Photic zone euxinia only occurred during OAE 2. SST kept increasing in the aftermath of OAE 2. The authors conclude that warming played a critical role in the spread of anoxia in the central Atlantic before, and during OAE 2.

**AC- The authors are grateful to the reviewer #1 for taking time to provide constructive comments for this manuscript. All the comments have been taken into consideration in the revised manuscript.**

**General comment**:
The manuscript is concise and clear and overall, very well constructed, and pleasant to read. New data are not really unexpected nor ground-breaking (since they overlap in part with van Bentum et al., 2009 and O'Brien et al., 2017), but they are interesting and new. Above all, I really appreciated the way the authors used previous work: the reader easily follows what data were generated in this study, what data come from previous work, and the authors compare their new data with previous work in a clever and accurate way. I write this review more rapidly than usual because I am asked to send it by May 1st, and I am currently at EGU, but that's fine with me since I have only minor comments and suggest prompt publication after minor revisions anyway (Please note that I'm no geochemist and am not able to evaluate the robustness of the geochemical analyses).

**Minor comments**:

**1**. Impact of the orbital configuration: The authors very rapidly approach the question of astronomical forcing (line 294). I think they could tell a bit more about that, notably in the light of key modelling (Sarr et al., 2019) and data (Laurin et al., 2016; Li et al., 2017) studies. I think that invoking the orbit:
• is important in the introduction.
• might help explaining the onset and/or termination of OAE 2 against a background of increasing SSTs.

**AC – We thank the reviewer for this comment. The manuscript primarily explores the long-term relationship between temperature and anoxia, but the variability in the record does prompt consideration of orbital impacts on oceanic anoxia during the Cretaceous period (Baternburg, 2016). We are reluctant to include this in the introduction, because that will somewhat misrepresent the data to come. We are also reluctant to overinterpret our data due to its resolution. However, we have added the suggested modelling references to the discussion to put our results into a wider context.**

**2**. Please check the figures:
• I think Fig. 1 is not called in the main text, but should be kept in the MS;

• I think Fig. 2 is not called and am not sure it's really useful;
• Fig. 3 is very nice but laterally very compressed, so that all temporal trends are difficult to read (e.g., the large SST increase, which looks like a flat line; or similarly: "TOC contents in excess of 5%" on 187: this is typically very difficult to see on the figure). Would that be possible to solve that problem? It would at least be helpful to draw vertical lines for key values (i.e., a figure 'grid'). Also, on line 141, the authors refer to some depth interval, but depth is not shown in Fig. 3. Please add it or convert depth to age in the text.

**AC- The authors agreed that the figures need some revision based on the comments. The figures are updated on the revised manuscript, incorporating all of these suggestions.**

**3**. Lines 146–147: "SSTs are lower than those of Foster et al (2007)". By how much? Again, this is difficult to estimate based on Fig. 3.

**AC- The SSTs are about 2.6 °C lower and that has been included in the revised manuscript.**

**4**. Lines 182–183: "from the Albian to the Cenomanian". Albian not shown?

**AC – We have corrected this mistake.**

**5**. Lines 226–227: Regarding sulfurization of OM during OAE 2, please consider the very nice quantification by (Hülse et al., 2019).

**AC – We have incorporated Hulse et al. into the discussion.**

**Technical comments**:
• Line 44: 'key biogeochemical cycles'
• Line 76 and throughout (e.g., lines 140, 142, 157): Fig. 3.1 should be Fig. 3 I think.
• Line 226: I guess "OC" stands for organic carbon, please define upon first use or remove the acronym (which I think is used here only).

**AC- All of these suggestions have been incorporated in the revised manuscript.**

**References cited**:
- Hülse, D., Arndt, S., Ridgwell, A., 2019. Mitigation of Extreme Ocean Anoxic Event Conditions by Organic Matter Sulfurization. Paleoceanography and Paleoclimatology 34, 476–489. https://doi.org/10.1029/2018PA003470
- Laurin, J., Meyers, S.R., Galeotti, S., Lanci, L., 2016. Frequency modulation reveals the phasing of orbital eccentricity during Cretaceous Oceanic Anoxic Event II and the Eocene hyperthermals. Earth and Planetary Science Letters 442, 143–156. https://doi.org/10.1016/j.epsl.2016.02.047
- Li, Y.-X., Montañez, I.P., Liu, Z., Ma, L., 2017. Astronomical constraints on global carbon-cycle perturbation during Oceanic Anoxic Event 2 (OAE2). Earth and Planetary Science Letters 462, 35–46. https://doi.org/10.1016/j.epsl.2017.01.007
- Sarr, A.C., Sepulchre, P., Husson, L., 2019. Impact of the Sunda Shelf on the Climate of the Maritime Continent. Journal of Geophysical Research: Atmospheres 124, 2574–2588. https://doi.org/10.1029/2018JD029971

---

## Author Comment (AC2)

**Responses to Reviewer #2's comments**

**General comments:**

This paper looks at the organic geochemistry of the black shales deposited in the run-up to OAE 2 at Demerara Rise in the equatorial proto-Atlantic. Using a range of biomarkers, the authors plot the increase in deoxygenation that moved in concert with increasing temperature, as documented by $TEX_{86}$ data. Although association does not prove cause and effect, the palaeoceanographic model they suggest makes general sense and they are careful to look also at the palaeotectonic context of their section in the light of the evolving South Atlantic, which could have impacted basin geometry and watermass stratification. An interesting highlight of the paper is the switch away from lycopane that is present in the upper Cenomanian to isorenieratane over the OAE 2 interval itself (Cenomanian–Turonian boundary), suggesting a change in the bacterioplankton consortium as photic-zone euxinic conditions took hold: presumably due to invasion by green sulfur bacteria.

**AC- The authors are thankful to the reviewer #2 for taking time to provide constructive comments for this manuscript. All the comments have been taken into consideration in the revised manuscript.**

**Specific comments:**

Given the sampling density of this core, and the fact that the sedimentary material can move around, I wondered whether the samples giving new data could be accurately fixed in the stratigraphy and combined with pre-existing data. Do the authors have any feeling for this?

**AC- The authors acknowledge the dynamic of the sedimentary material presented in this manuscript. Although there could be slight shifts in the absolute depth of the two datasets relative to each other, we are confident that the data used in the manuscript are in the correct stratigraphic order based on discussions with authors of previous studies and isotopic comparisons.**

My main grouse in the account is the mixing between rock (or sediment) and time and between rock and process. You cannot sample an OAE or pass up into it: it is a phenomenon that leaves a distinctive record. You cannot have a TOC value for OAE 2! Use of the term 'interval', which can be applied to both sediment and time, can be helpful.

Geological narrative should not be in the present tense.

In-text references should be preferably ordered consistently by date

**AC- That is a wise observation. We acknowledge that we have conflated the OAE interval with the sediments at this site, and that is incorrect. In the revised manuscript, we have followed the advice to use the term 'OAE 2 interval'. We have also revised the tenses and references as based on the comments.**

Details:

Line 23:  I would hyphenate 'water column' where used as a compound noun–noun adjective.
**Done**

Line 27 and elsewhere: although the journal allows alphabetical in-text citations, I think that ordering by date is much to be preferred, as this technique indicates the academic trajectory of the point in question and gives credit where it is due.   Referencing is date ordered in some places in the manuscript . . . . but see below (Line 31 and 34).  Make consistent throughout the manuscript.
**Done**

Line 31: references are not ordered by either alphabet or date!
**Done**

Line 34: references are not ordered by either alphabet or date!
**Done**

Line 41:  Avoid beginning a sentence with an unqualified 'This' or 'These', which is often ambiguous.  I suggest 'These phenomena'
**Done**

Line 76: I assume 'sediment' should be 'samples'?  And change 'was' to 'were'
**Done**

Line 77: better would be "Cenomanian to Turonian interval'
**Done**

Line 79: here you are mixing rock and time.  Unit IV is stratigraphically underlain by Albian phosphatic calcareous claystone
**Done**

Line 83: there is confusion here between a phenomenon (such as an OAE) and sediment.  You cannot sample an OAE, only its sedimentary record.  Rephrase with something like 'over the OAE 2 interval'. ('Interval' can be used for both rock and time and is a very useful term in this regard).  I suggest changing 'following' (time word) to post-dating.
**Done**

Line 84: you cannot identify OAE 2 at Site 1258 (it is long gone!)  Change to 'OAE 2 interval'
**Done**

Line 101? We know that sediments can move position in a core, so conflating data sets from samples taken years apart can be dangerous if correct stratigraphic order is not retained.  How confident are the authors that all data are in correct stratigraphic order?
**AC- The authors confirm the correct stratigraphic order for all the samples involved using the repository samples identification from MARUM, where the cores were kept. Then, the authors acquired the samples list from Erbacher for the published data and confirmed that samples used were within the interval and isotopically ($\delta^{13}C$)**

**superimposed to new data (GDGT samples-SST proxy) presented in this study. The average spacing for isotopes samples and GDGT samples are 0.12 m and 0.74 m, respectively. This indicates at least 6 samples corresponding data points in 0.12m dataset. Moreover, these two datasets indicate a very strong linear correlation between these two datasets ($r^2$=0.9958). Hence, the stratigraphic order is robust.**

Line 117: better would be '...serves as a proxy for water-column anoxia. . . .'
**Done**

Line 142: 'during' is a time word and you are describing a geochemical characteristic of a sediment sample. ' . . in the OAE 2 interval . . . ' would be better
**Done**

Line 158" change to 'water-column' (with a hyphen)
**Done**

Line 162: change to 'up to the OAE 2 intervaL'
**Done**

Line 166: change to 'lower Cenomanian': this is rock not time.
**Done**

Line 167: water-column
**Done**

Line 177: replace ' "Following OAE 2' by 'stratigraphically higher than the OAE 2 interval' or similar.
**Done**

Line 178: change to 'Upper Cretaceous' – you are describing a feature of the sediment
**Done**

Line 182: delete hyphen after 'long-'
**Done**

Line 183" better would be: 'culminating in the OAE 2 interval
**Done**

Line 190: hyphenate 'bottom water' used as a compound adjective
**Done**

Line 191: better would be ' just below the onset level of OAE 2' (to avoid mixing sediment with a phenomenon)
**Done**

Line 194: replace 'sometime' with 'in some cases'
**Done**

Line 205" 'over the OAE 2 interval'
**Done**

Line 203: but presumably impinging on the sea floor if benthic foraminifera are affected?
**Done**

Line 205: change to 'This phenomenon has also been reported. . . .'
**Done**

Line 211: change to 'highest in the OAE 2 interval . . . '
**Done**

Line 211: change to 'were driven' – this is geological narrative
**Done**

Line 213" 'were replaced'
**Done**

Line 218: 'over the OAE 2 interval' would be better
**Done**

Line 219: water-column
**Done**

Line 224: change 'into' to 'during'
**Done**

Line 225: water-column
**Done**

Line 226: This process also gives rise. . . .
**Done**

Line 230/231: change to : 'water-column'
**Done**

Line 234: change to:  'these metal-isotope data by confirming  . . .'
**Done**

Line 234: water-column
**Done**

Line 235: change 'adding' to 'adds'
**Done**

Line 235 Line 225: water-column
**Done**

Line 240: delete hyphen from adverb to give 'nearly closed' – this is journal house style
**Done**

Line 244: change to 'This result. . . .
**Done**

Line 245: change 'of' to 'for'
**Done**

Line 347: Line 225: water-column
**Done**

Line 253: 'Therefore, this result illustrates the occurrence of marine water-column anoxia.
**Done**

Line 256: change to 'propagated'. .'
**Done**

Line 257: should this be 'bottom-water anoxia'?
**Done**

Line 257: change to 'was linked'
**Done**

Line 258: change to 'was overridden'
**Done**

Line 259: 'watermass' (one word)
**Done**

Line 260: change to 'in contrast to. . . .'
**Done**

Line 264: should this be 'deep-water convection'?
**Done**

Line 266: water-column
**Done**

Line 271: change to 'during the Cenomanian. . . .'
**Done**

Line 273: change to 'during this time' (not necessary to repeat 'Cenomanian'
**Done**

Line 284: change to 'that persisted. . . .'
**Done**

Line 287: change ''is not' to 'was not'
**Done**

Line 289: change to 'was related'
**Done**

Line 292: water-column
**Done**

Line 295: not clear what 'these' refers to – clarify
**Done**

Line 298: water-column
**Done**

Line 298/299: anoxia correlates with warming but association does not prove cause and effect
**Done**

Line 300: water-column
**Done**

Line 305: better would be 'albeit modulated by climatic factors
**Done**

579/580" Cross-plot and water-column (add hyhens)
**Done**

**AC- All technical comments (Line 23 to 580) are corrected in the revised manuscript.**